# A single full-length VAR2CSA ectodomain variant purifies broadly neutralizing antibodies against placental malaria isolates

Justin YA Doritchamou[1], Jonathan P Renn[1], Bethany Jenkins[1], Almahamoudou Mahamar[2], Alassane Dicko[2], Michal Fried[1], Patrick E Duffy[1]*

[1]Laboratory of Malaria Immunology and Vaccinology, National Institute of Allergy and Infectious Diseases, National Institutes of Health, Bethesda, United States; [2]Malaria Research and Training Center, University of Sciences, Techniques, and Technologies of Bamako, Bamako, Mali

*For correspondence:
patrick.duffy@nih.gov

Competing interest: The authors declare that no competing interests exist.

**Abstract** Placental malaria (PM) is a deadly syndrome most frequent and severe in first pregnancies. PM results from accumulation of *Plasmodium falciparum*-infected erythrocytes (IE) that express the surface antigen VAR2CSA and bind to chondroitin sulfate A (CSA) in the placenta. Women become PM-resistant over successive pregnancies as they develop anti-adhesion and anti-VAR2CSA antibodies, supporting VAR2CSA as the leading PM-vaccine candidate. However, the first VAR2CSA subunit vaccines failed to induce broadly neutralizing antibody and it is known that naturally acquired antibodies target both variant-specific and conserved epitopes. It is crucial to determine whether effective vaccines will require incorporation of many or only a single VAR2CSA variants. Here, IgG from multigravidae was sequentially purified on five full-length VAR2CSA ectodomain variants, thereby depleting IgG reactivity to each. The five VAR2CSA variants purified ~0.7% of total IgG and yielded both strain-transcending and strain-specific reactivity to VAR2CSA and IE-surface antigen. In two independent antibody purification/depletion experiments with permutated order of VAR2CSA variants, IgG purified on the first VAR2CSA antigen displayed broad cross-reactivity to both recombinant and native VAR2CSA variants, and inhibited binding of all isolates to CSA. IgG remaining after depletion on all variants showed significantly reduced binding-inhibition activity compared to initial total IgG. These findings demonstrate that a single VAR2CSA ectodomain variant displays conserved epitopes that are targeted by neutralizing (or binding-inhibitory) antibodies shared by multiple parasite strains, including maternal isolates. This suggests that a broadly effective PM-vaccine can be achieved with a limited number of VAR2CSA variants.

## Introduction

*Plasmodium falciparum* infection in pregnant women causes placental malaria (PM) when *P. falciparum*-infected erythrocytes (IE) accumulate in the intervillous spaces of the placenta. PM has been linked to several adverse pregnancy outcomes (*Steketee et al., 2001*; *Guyatt and Snow, 2001*; *Desai et al., 2007*; *Moore et al., 2017*), and first-time mothers are most vulnerable (*Brabin, 1983*). Over successive pregnancies, PM and the related sequalae become less prevalent (*Guyatt and Snow, 2001*). Susceptibility to PM has been attributed to *P. falciparum* parasites that bind chondroitin sulphate-A (CSA) expressed by the placental syncytiotrophoblast (*Fried and Duffy, 1996*), and express the variant surface antigen VAR2CSA (*Salanti et al., 2003*; *Tuikue Ndam et al., 2005*). Conversely, the decrease in PM-related poor pregnancy outcomes with increasing parity is associated with the acquisition of

**eLife digest** Contracting malaria during pregnancy – especially a first pregnancy – can lead to a severe, placental form of the disease that is often fatal. Red blood cells infected with the malaria parasite *Plasmodium falciparum* display a protein, VAR2CSA, which can recognize and bind CSA molecules present on placental cells and in placental blood spaces. This leads to the infected blood cells accumulating in the placenta and inducing harmful inflammation.

Having been exposed to the parasite in prior pregnancies generates antibodies that target VAR2CSA, stopping the infected blood cells from latching onto placental CSA or tagging them for immune destruction. Overall, this makes placental malaria less severe in following pregnancies, and suggests that vaccines could be developed based on VAR2CSA.

However, this protein has regions that can vary in structure, meaning that *P. falciparaum* can generate many VAR2CSA variants. Individuals exposed to the parasite naturally generate antibodies that block a wide array of variants from attaching to CSA. In contrast, first-generation vaccines based on VAR2CSA fragments have only induced variant-specific antibodies, therefore offering limited protection against infection.

As a response, Doritchamou et al. set out to find VAR2CSA structures that could be recognized by antibodies targeting an array of variants. Blood was obtained from women who had had multiple pregnancies and were immune to malaria. Their plasma was passed over five different large VAR2CSA variants in order to isolate and purify antibodies that attached to these structures.

Doritchamou et al. found that antibodies binding to individual VAR2CSA structures could also recognise a wide array of VAR2CSA variants and blocked all tested parasites from sticking to CSA. While further research is needed, these findings highlight antibodies that cross-react to diverse VAR2CSA variants and could be used to design more effective vaccines targeting placental malaria.

functional antibodies to CSA-binding IE (*Fried and Duffy, 1998*; *Ricke et al., 2000*) and antibodies to VAR2CSA (*Salanti et al., 2004*; *Ndam et al., 2015*). Such functional antibodies have been characterized for two major functions: (1) blocking CSA-binding of VAR2CSA-expressing parasites and (2) opsonizing IE to promote phagocytosis (*Fried and Duffy, 1998*; *Ricke et al., 2000*; *Duffy and Fried, 2003*; *Keen et al., 2007*; *Ataíde et al., 2011*).

Hence, VAR2CSA represents the leading candidate for PM vaccine development. VAR2CSA is a large (~318–478 kDa) multidomain transmembrane protein, a member of the *P. falciparum* erythrocyte membrane protein 1 (*Pf*EMP1) family encoded by *var* genes (*Gardner et al., 2002*; *Hviid and Jensen, 2015*). The cysteine-rich ectodomain is formed by N-terminal sequence (NTS), six and sometimes more Duffy-binding-like (DBL) domains as well as interdomain (ID) regions (*Kraemer and Smith, 2006*; *Doritchamou et al., 2019*). Recent studies showed that VAR2CSA ectodomain structure includes a stable core (NTS-DBL1X-ID1-DBL2X-ID2-DBL3X-DBL4e-ID3) flanked by a flexible arm (DBL5e -DBL6e), and the receptor interaction involves CSA threading through two channels that formed within the stable core (*Ma et al., 2021*; *Wang et al., 2021*). Multiple individual DBL domains of VAR2CSA interact with CSA in vitro (*Dahlbäck et al., 2011*; *Clausen et al., 2012*; *Ma et al., 2021*) and induce functional antibodies in animals (*Bigey et al., 2011*; *Fried et al., 2013*; *Nielsen and Salanti, 2015*; *Chêne et al., 2018*). Two subunit vaccine candidates (called PAMVAC and PRIMVAC) from the N-terminal region of VAR2CSA were recently tested in phase one trials (*Mordmüller et al., 2019*; *Sirima et al., 2020*). Both trial teams reported that VAR2CSA subunit vaccines were safe, immunogenic and induced functional anti-adhesion antibodies in malaria-naive and malaria-exposed women (*Mordmüller et al., 2019*; *Sirima et al., 2020*). However, functional activity was primarily against homologous parasites (same VAR2CSA sequence as vaccine) and low or absent against heterologous parasites (*Sirima et al., 2020*), suggesting the vaccine antigen failed to display functional epitopes shared across multiple *P. falciparum* variants (reviewed in *Doritchamou et al., 2021*).

Similarly, VAR2CSA fragments (single or double domains) fail to purify the broadly neutralizing activity of sera from PM-resistant multigravidae (*Doritchamou et al., 2016*). Although VAR2CSA fragments (including constructs similar to PAMVAC vaccine) purified antibodies with CSA-binding inhibitory activity against homologous parasites, broadly neutralizing activity was neither purified nor depleted after passing sera over several VAR2CSA domains and variants (*Doritchamou et al., 2016*).

These data supports the hypothesis that functional antibodies to VAR2CSA may target conformational epitopes not displayed by any individual domain: for example, the strain-transcending human monoclonal antibody PAM1.4 is specific for an unknown conformational epitope on full-length VAR2CSA (*Barfod et al., 2007*).

Naturally acquired antibodies of multiparous women exhibit broadly neutralizing anti-adhesion activity (*Fried and Duffy, 1998*; *Ricke et al., 2000*; *Ndam et al., 2015*). However, it is unknown whether broadly binding-inhibitory antibodies target conserved epitopes on VAR2CSA or rather result from the cumulative repertoire of antibodies against variant-specific epitopes. In this study, we investigate the hypothesis that full-length VAR2CSA ectodomain may capture naturally acquired antibodies from multigravidae plasma with greater breadth of neutralizing activity than previously achieved by multiple VAR2CSA domains. We show that a single ectodomain can purify strain-transcending IgG as well as the bulk of functional activity naturally acquired by multigravid women. In parallel, the depletion of IgG reactivities to five variants of full-length VAR2CSA resulted in significantly diminished strain-transcending anti-adhesion activity. These data suggest that full-length VAR2CSA ectodomain

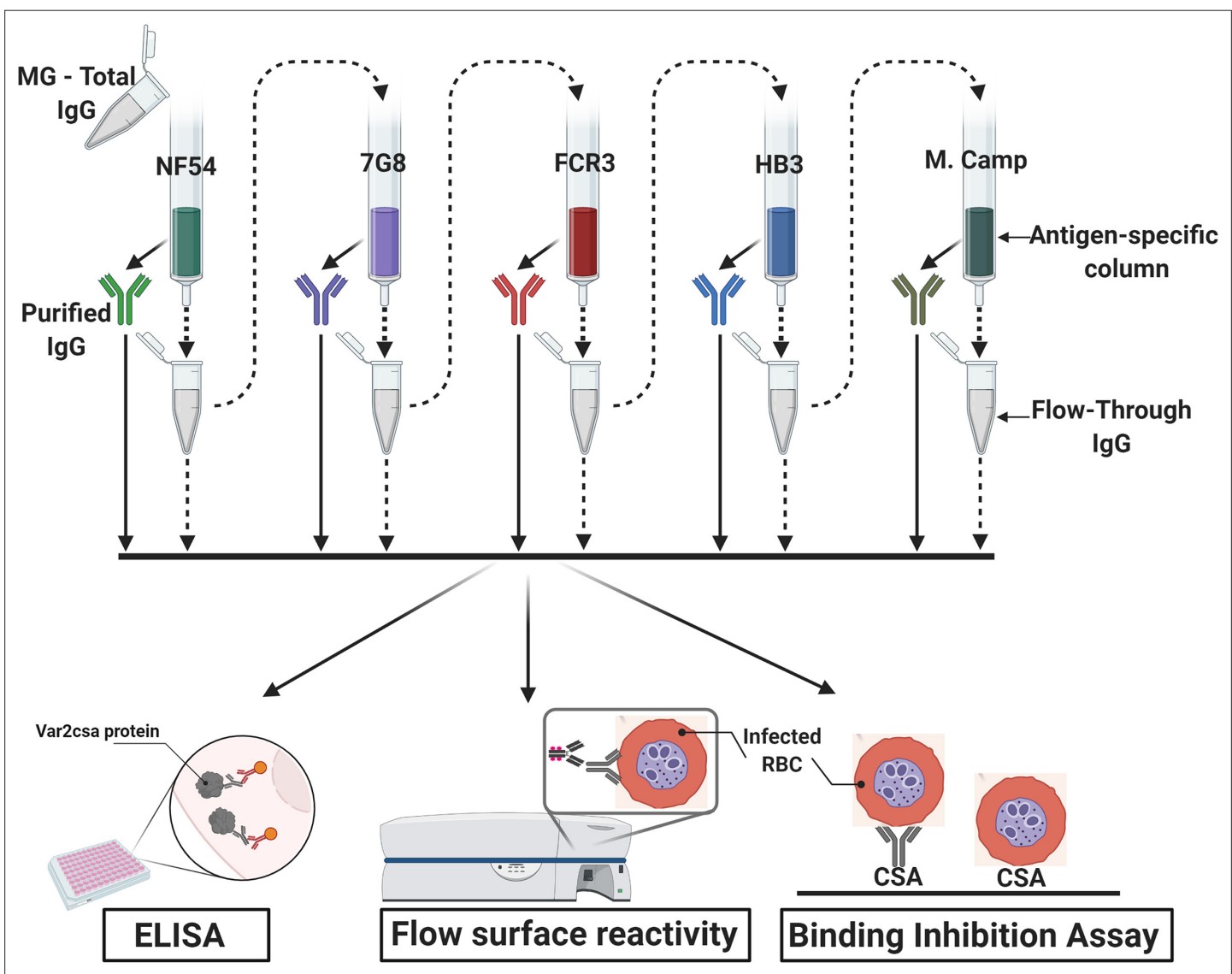

**Figure 1.** Flow chart of the experimental design. VAR2CSA-specific IgG was sequentially purified from total IgG isolated in multigravidae (MG) using antigen-specific columns made with NF54, 7G8, FCR3, HB3, Malayan Camp (M. Camp) alleles of full-length VAR2CSA ectodomain. The flow-through IgG as well as eluted IgG after each VAR2CSA column were assessed by ELISA, in Flow cytometry for surface reactivity and binding inhibition assay using CSA-binding *Plasmodium falciparum*-infected red blood cells (RBC). Image was created with https://BioRender.com.

**Table 1.** Purification yields of purified IgG.

| Order of depletion | IgG | Yield |
|---|---|---|
| - | Total IgG | 58,000 µg |
| 1 | Fv2-NF54 IgG | 145.6 µg |
| 2 | Fv2-7G8 IgG | 148.9 µg |
| 3 | Fv2-FCR3 IgG | 53.4 µg |
| 4 | Fv2-HB3 IgG | 14.5 µg |
| 5 | Fv2-M. Camp IgG | 23.7 µg |

displays functional epitopes that are absent in subunit VAR2CSA fragments and provide a basis to design improved vaccines.

# Results

## Depletion of IgG on five VAR2CSA ectodomains significantly reduces broad neutralizing activity

IgG specific to full-length VAR2CSA ectodomains was purified from a plasma pool prepared using samples from multigravid women participating in the previously described Immuno-epidemiology (IMEP) study (*Fried et al., 2018*) (see Materials and methods and *Figure 1* for experimental design).

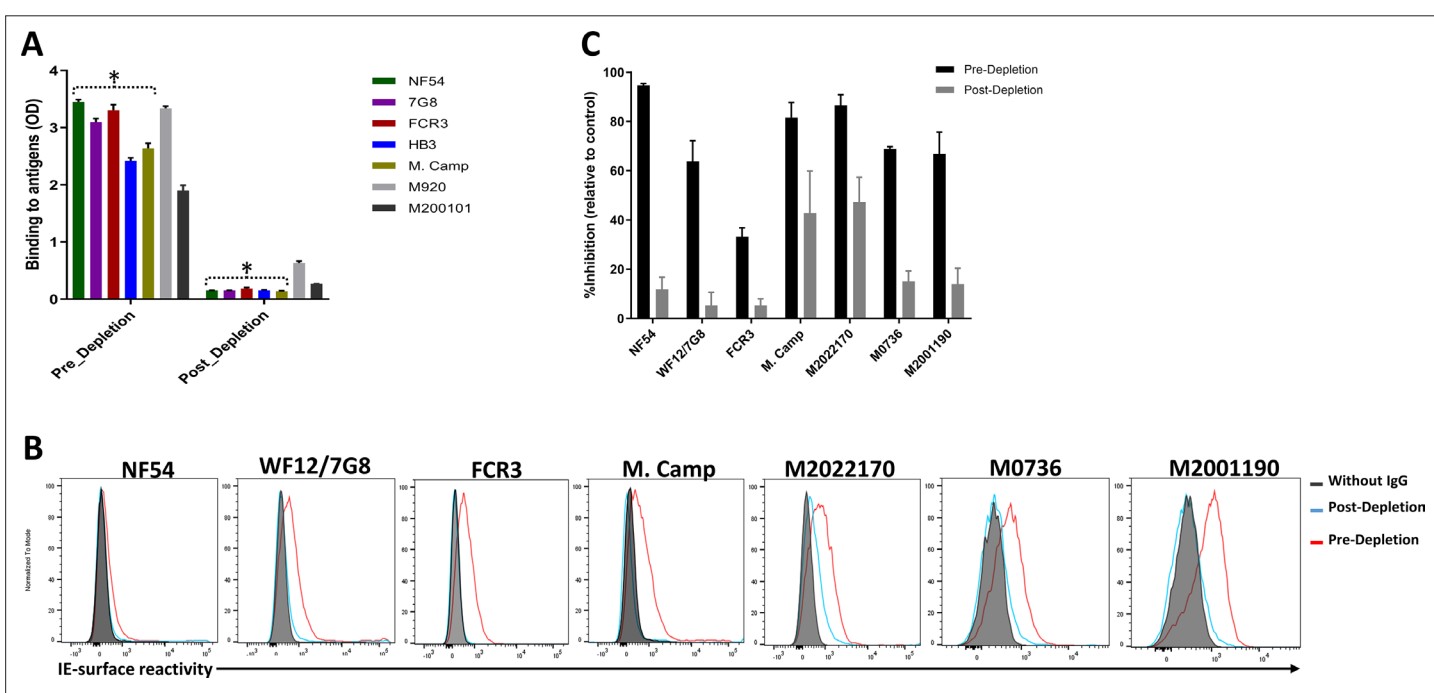

**Figure 2.** Depletion of VAR2CSA-specific IgG and activity of Total IgG purified from Malian multigravidae. Activity of the total IgG before (Pre-depletion) and after (Post-depletion) purification of VAR2CSA-specific IgG was assessed by (**A**) ELISA on different recombinant full-length VAR2CSA ectodomain, (**B**) by Flow cytometry, and (**C**) CSA-binding inhibition assay using six isolates including three recently adapted maternal isolates (M0736, M2022170, and M2001190). In panel B, histogram without IgG represents assay well with no testing IgG and similarly stained with conjugated anti-human secondary IgG. In panel C, percent inhibition activity of purified IgG was determined relative to activity obtained with control wells without any testing IgG. Data in C represent at least two independent experiments. Asterisk (*) indicates antigens used in the sequential VAR2CSA-specific IgG depletion assay.

The online version of this article includes the following source data and figure supplement(s) for figure 2:

**Source data 1.** ELISA activity of total IgG before and after purification of VAR2CSA-specific IgG.

**Source data 2.** CSA-binding inhibition activities before and after purification of VAR2CSA-specific IgG.

**Figure supplement 1.** Reactivity of the post depletion samples to VAR2CSA recombinants and CSA-binding isolates.

**Figure supplement 2.** VAR2CSA sequences shared by progeny of 7G8 and GB4 **crossing**.

**Figure supplement 3.** Percent reduction in CSA-binding inhibition activity of the MG-Pool after VAR2CSA IgG purification/depletion.

**Figure supplement 3—source data 1.** Percent reduction in CSA-binding inhibition activity of the MG-Pool after VAR2CSA IgG purification/depletion.

**Figure supplement 4.** Phylogenetic analysis of variants of full-length VAR2CSA sequences.

The affinity purification of IgG specific to five VAR2CSA ectodomain variants yielded a combined ~386.1 μg of VAR2CSA-specific IgG, representing about 0.7% of the original total IgG (*Table 1*). The highest yields of purified IgG were obtained on the first two variants (NF54 and 7G8) totaling 76% of the IgG purified on all five variants.

Variable levels of IgG reactivity to seven recombinant VAR2CSA ectodomains was measured in the pre-depletion total IgG. As expected, IgG reactivity to NF54, 7G8, FCR3, HB3, and M. Camp variants were successfully depleted from the total IgG pool, as demonstrated by the absence of reactivity in each post-depletion sample (*Figure 2A*, *Figure 2—figure supplement 1*). Of note, substantially reduced reactivity to M920 and M200101 (not used in the affinity purification assay) was seen in the post-depletion total IgG, indicating that cross-reactive IgG to these two antigens was also depleted.

Similarly, reactivity to native VAR2CSA measured on the IE surface by flow cytometry was depleted in the post-depletion IgG when tested on NF54, WF12/7G8, FCR3 and Malayan Camp (M. Camp) isolates (*Figure 2B*, *Figure 2—figure supplement 1*). Of note, WF12 (here named WF12/7G8) is a progeny of 7G8 X GB4 (*Hayton et al., 2008*) and its genome encodes the 7G8 VAR2CSA variant (*Figure 2—figure supplement 2*) Further, reactivity of post-depletion total IgG to native VAR2CSA of heterologous maternal isolates (M2022170, M2001190, and M0736) was substantially reduced versus the pre-depletion total IgG.

Importantly, depletion of IgG on the five VAR2CSA variants also reduced CSA-binding inhibition activity in the total IgG, being abolished against some isolates and significantly reduced against others

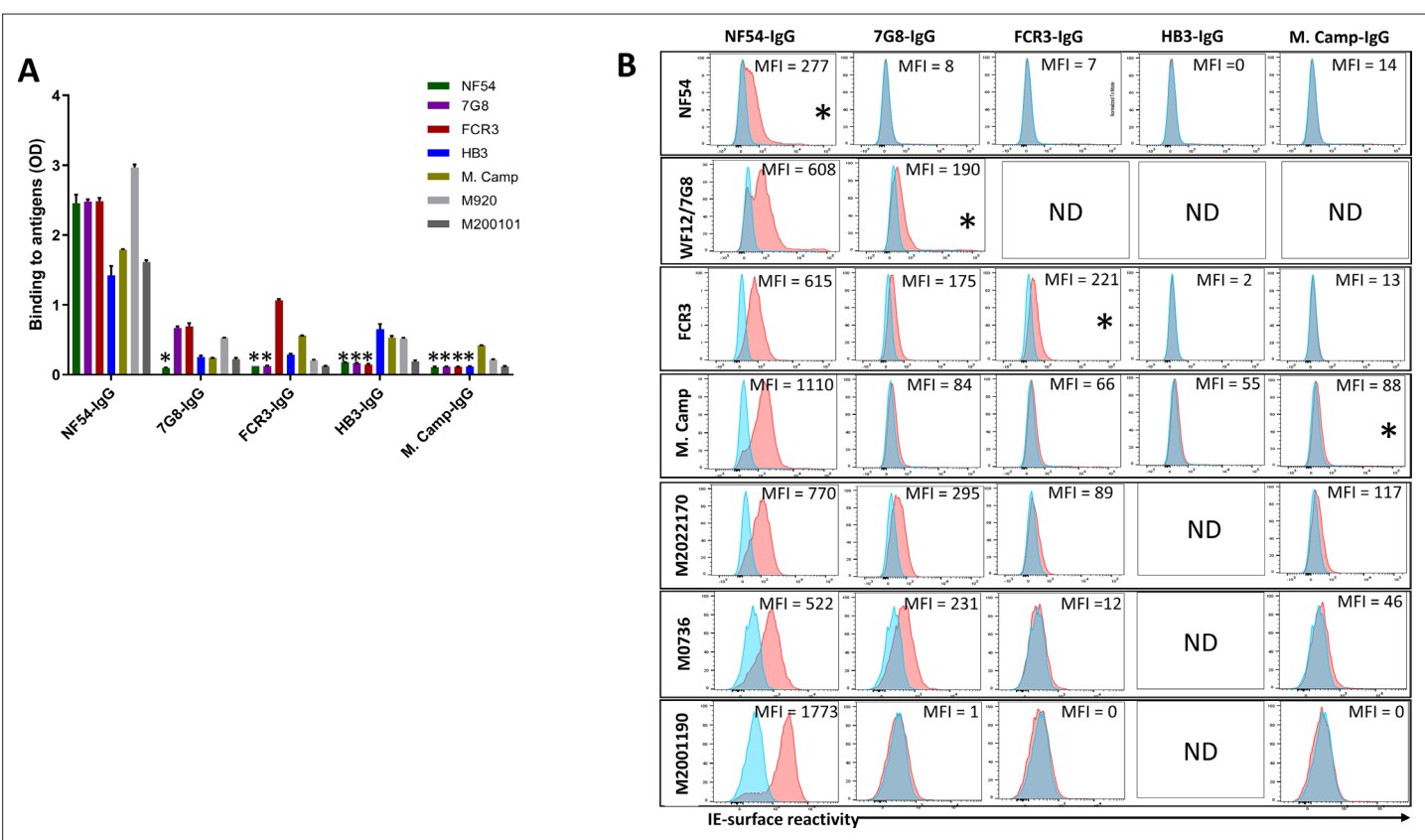

**Figure 3.** Reactivity of the purified VAR2CSA-specific IgG to different full-length VAR2CSA antigens and *P. falciparum* isolates. Reactivity of VAR2CSA-specific purified IgG was assessed by ELISA (**A**) on seven recombinant full-length VAR2CSA ectodomain proteins including two proteins (M920 and M200101) that were not used in the purification assay. IgG were tested at 0.1 μg/mL concentration (IgG conc.); asterisk (*) indicates depleted reactivity. (**B**) Ability of purified VAR2CSA IgG to bind native VAR2CSA on six CSA-binding isolates was evaluated by Flow cytometry. Testing IgG is indicated in red while histogram in blue represents assay well with no testing IgG and similarly stained with conjugated anti-human secondary IgG. Median fluorescence intensity (MFI) values are indicated. Due to limited material of purified IgG on HB3 VAR2CSA (HB3-IgG), no data (ND) are available for maternal isolates (M0736, M2022170 and M2001190). Asterisk (*) indicates surface reactivity to the homologous parasite.

The online version of this article includes the following source data for figure 3:

**Source data 1.** ELISA reactivity of VAR2CSA-specific purified IgG on 7 recombinant full-length VAR2CSA ectodomain proteins.

(*Figure 2C*). The inhibition activity of the original pre-depletion total IgG sample was >65% against all parasite lines including recently adapted maternal isolates, except for FCR3 whose binding was inhibited 33% (*Figure 2C*). After the depletion of VAR2CSA-specific IgG, the inhibition activity was substantially reduced, with the % reduction (measured as (pre-depletion % inhibition-post-depletion % inhibition)/(pre-depletion % inhibition) x 100) ranging from 45% against M2022170% to 87% against NF54 (*Figure 2—figure supplement 3*). CSA-binding inhibition activity was substantially reduced against the three maternal isolates (including M2022170), implying that a significant fraction of cross-inhibitory IgG to the maternal isolates were also depleted on the heterologous VAR2CSA antigens.

## Highly cross-reactive IgG is purified on the first full-length VAR2CSA antigen

VAR2CSA-purified IgG exhibited cross-reactivity in ELISA to different variants including M920 and M200101 (*Figure 3A*), although the degree of this dropped substantially after the first purification. As expected, reactivity was largely absent when IgG was tested against VAR2CSA variants previously used in the series of purifications/depletions. Thus, NF54-purified IgG (NF54-IgG) reacted to homologous and heterologous VAR2CSA antigens at the highest level of any IgG, while IgG purified on M. Camp (the last antigen used in the purification/depletion series) showed no cross-reactivity to NF54, 7G8, FCR3, or HB3 variants. Notably, each variant purified specific reactivity to itself, albeit at lower OD values than that seen with the initial NF54 purification, which may indicate the greater abundance and/or greater affinity of cross-reactive versus variant-specific IgG acquired by multigravidae.

Similar to ELISA, flow-cytometry revealed strain-transcending reactivity to native antigen, although this decreased after the first (NF54) and again after the second (7G8) affinity columns. NF54-IgG showed the greatest breadth of reactivity (defined as the number of isolates recognized) with detectable reactivity to IE surface of all parasites including the maternal isolates (known to be polyclonal parasite samples). NF54-IgG also displayed the highest level of reactivity to all the isolates (*Figure 3B*). Compared to NF54-IgG, IgG subsequently purified on 7G8, FCR3, and M. Camp ectodomains demonstrated lower surface reactivity to the homologous parasite and lower cross-reactivity

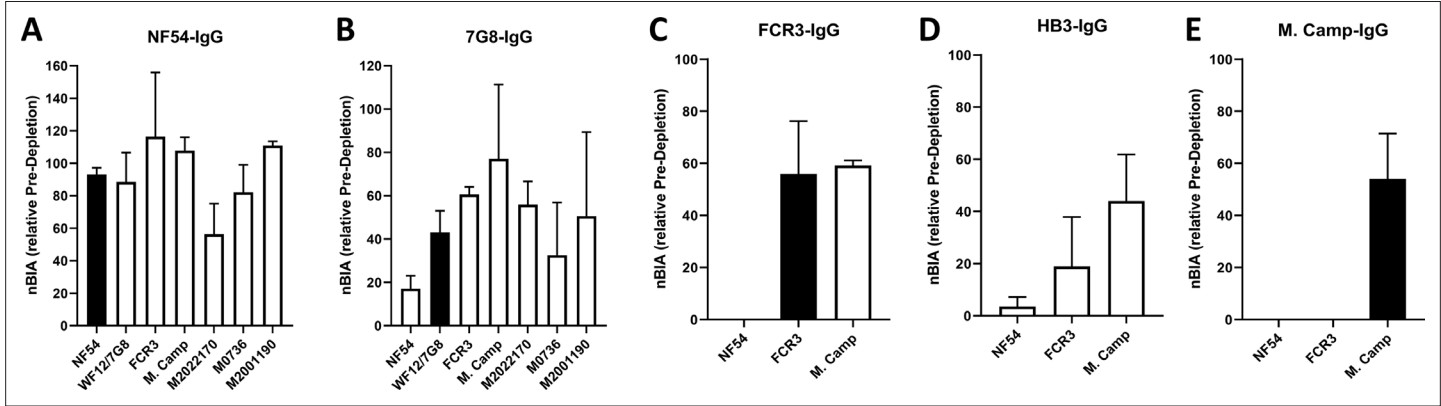

**Figure 4.** CSA-binding inhibitory activity of VAR2CSA-specific purified IgG from multigravidae. The inhibition capacity of VAR2CSA-specific purified IgG was assessed on CSA-binding isolates (A – E). Blocking activity (nBIA) of purified IgG was normalized by activity obtained with total IgG before any VAR2CSA IgG purification (Pre-depletion). Due to limited material, purified IgG on FCR3, HB3, and M. Camp was not tested on maternal isolates (M0736, M2022170, and M2001190). Black box indicates inhibition activity against homologous parasite and empty box represents activity against heterologous parasites. Data represent at least two independent experiments.

The online version of this article includes the following source data and figure supplement(s) for figure 4:

**Source data 1.** Inhibition capacity of VAR2CSA-specific purified IgG against NF54.

**Source data 2.** Inhibition capacity of VAR2CSA-specific purified IgG against 7G8.

**Source data 3.** Inhibition capacity of VAR2CSA-specific purified IgG against FCR3.

**Source data 4.** Inhibition capacity of VAR2CSA-specific purified IgG against HB3.

**Source data 5.** Inhibition capacity of VAR2CSA-specific purified IgG against M. Camp.

**Figure supplement 1.** CSA-binding inhibitory activity of VAR2CSA-specific IgG purified from multigravidae.

**Figure supplement 2.** Titration of homologous inhibition activity by purified VAR2CSA$_{NF54}$ specific IgG.

to the maternal isolates. Although the HB3-IgG was not tested against its homologous parasite line, this antigen-specific purified IgG displayed weaker cross-reactivity to different heterologous lab strain parasites compared to the other purified IgG. As with ELISA, reactivity by flow cytometry was lost against parasite isolates after their corresponding recombinant ectodomain had been used in the series of depletion/purifications, attesting to the complete depletion of homologous IE-surface reactivity by each homologous antigen.

## A single full-length VAR2CSA variant purifies the bulk of naturally acquired binding-inhibitory IgG

Strong cross-inhibitory activity of IgG purified on VAR2CSA was observed in assays that measure inhibition of parasite binding to CSA (*Figure 4—figure supplement 1*). IgG binding-inhibition activity was normalized (nBIA) as a percentage of the pre-depletion total IgG, with the assumption that maximum inhibition of each parasite line is measured in IgG before depletion. NF54-IgG contained the largest fraction of the inhibitory activity, with mean nBIA values > 50% against all isolates and hovering around 100% for most (*Figure 4A*). This included three recently adapted heterologous maternal isolates, which showed both the lowest (M2022170) and highest (M2001190) nBIA levels respectively at 56% and 111% inhibition. Interestingly, titration of the inhibitory capacity of NF54-IgG against the homologous parasite indicated that blocking is achieved at IgG concentration as low as 10 µg/mL (*Figure 4—figure supplement 2*). 7G8-IgG showed lower levels of nBIA than NF54-IgG against all heterologous isolates, except against the maternal isolate M2022170 where 7G8-igG showed similar levels of nBIA to NF54-IgG; as expected 7G8-IgG (and subsequent variant-purified IgG) showed essentially no activity against NF54 parasites (*Figure 4B–E*). Subsequent purifications of IgG on ectodomain variants continued to yield some level of homologous activity, as seen with WF12/7G8, FCR3, and M. Camp (*Figure 4C and E*). Taken together, the IgG purified on VAR2CSA variants clearly demonstrated homologous inhibition activity while displaying high levels of cross-inhibition activity with initial purifications that generally waned with successive purifications. Despite significant heterologous IE surface reactivities captured on the NF54 antigen, subsequent IgG purifications on other variants captured some level blocking activity against both homologous and heterologous parasites (generally at a lower level than that of NF54-IgG).

To confirm our finding that IgG purified on the first VAR2CSA variant displays the highest level of activities against various VAR2CSA-expressing isolates, we permutated the order of antigens in a second sequential purification/depletion experiment, where IgG to M. Camp, FCR3, and NF54 variants of full-length VAR2CSA were sequentially purified (*Figure 5—figure supplement 1*). IgG purified on the initial variant (M. Camp) exhibited the highest cross-reactivity to heterologous variants of recombinant VAR2CSA (*Figure 5A*) and native antigen expressed by CSA-binding isolates (*Figure 5B*). In the functional assay, M. Camp-IgG strongly cross-inhibited the binding of M. Camp, FCR3 and NF54 parasites to CSA (*Figure 5C, D and E*). This observation is consistent with the pattern observed with IgG purified on NF54 variant of VAR2CSA in the first purification/depletion experiment.

## Discussion

VAR2CSA is the leading vaccine candidate to protect malaria-exposed pregnant women against PM and related adverse outcomes. Two vaccines based on VAR2CSA fragments have been recently tested in clinical trials (*Mordmüller et al., 2019*; *Sirima et al., 2020*), showing good safety and immunogenicity but a general inability to induce broadly neutralizing antibodies, such as those described in sera of PM-resistant multigravid women (*Fried and Duffy, 1998*; *Ricke et al., 2000*; *Duffy and Fried, 2003*; *Ndam et al., 2015*). The limitations of VAR2CSA domain-based subunit vaccines have also been established from the studies of naturally acquired antibodies in PM-resistant multigravidae, where single domains or domain combinations from different variants failed to deplete heterologous binding-inhibition activity (*Doritchamou et al., 2016*). Among the probable explanations, the restricted array of protective epitopes displayed by VAR2CSA subunit vaccines, as well as the contribution of antibodies targeting other variants or antigens (including non-VAR2CSA proteins), are of major interest. It is therefore likely that larger fragments of VAR2CSA would offer a broader spectrum of functional epitopes including conformational epitopes and those present in other fragments missing from the subunits tested. Supporting this hypothesis, the present data demonstrate that

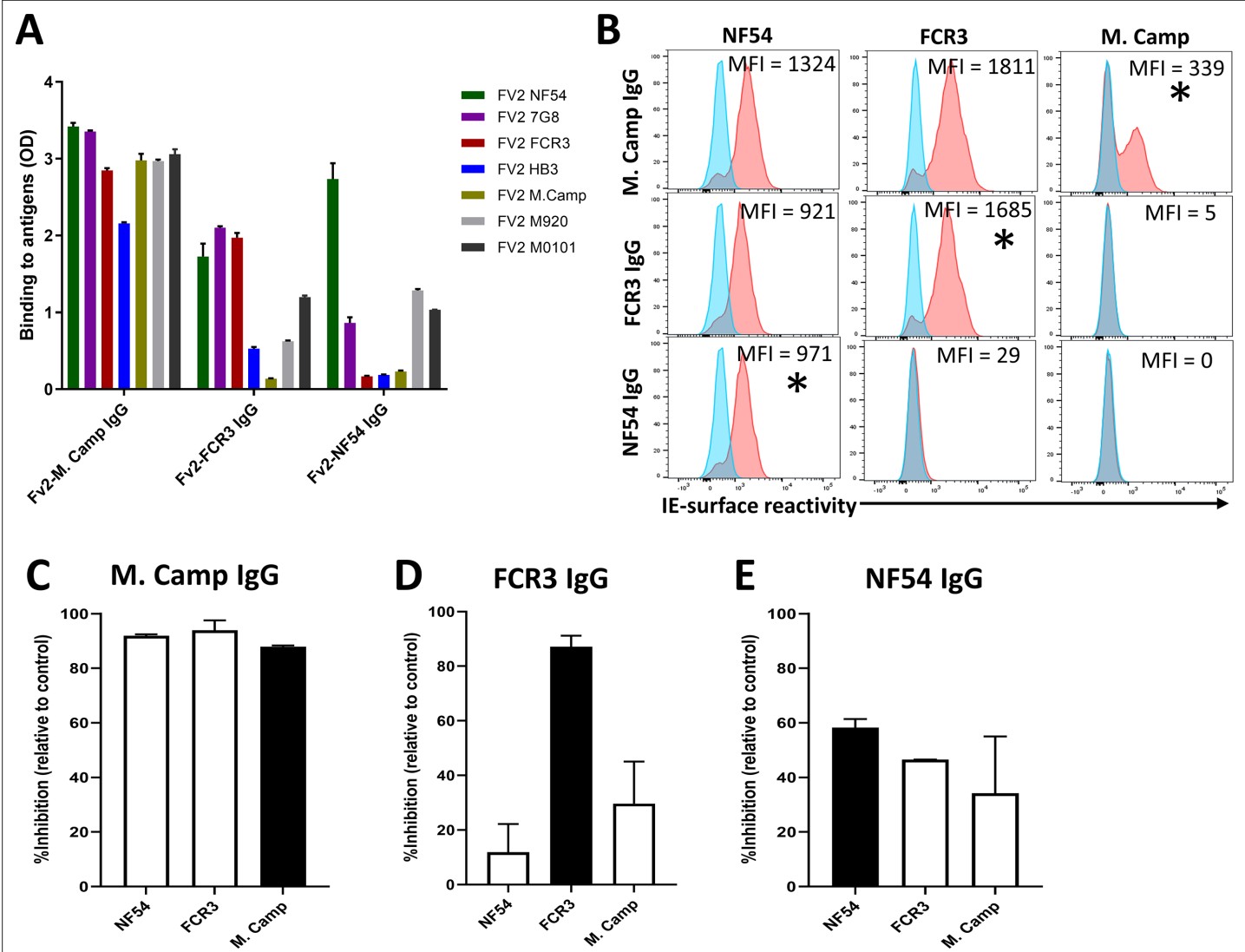

**Figure 5.** Reactivity of the purified VAR2CSA-specific IgG to different full-length VAR2CSA antigens and *P. falciparum* isolates in a second independent purification/depletion experiment. Reactivity of VAR2CSA-specific purified IgG was assessed by ELISA (**A**) on seven recombinant full-length VAR2CSA ectodomain proteins including four proteins (7G8, HB3, M920, and M200101) that were not used in this purification/depletion assay. IgG were tested at 0.1 µg/mL concentration. (**B**) Ability of purified VAR2CSA IgG to bind native VAR2CSA on six CSA-binding isolates was evaluated by Flow cytometry. Testing IgG is indicated in red while histogram in blue represents assay well with no testing IgG and similarly stained with conjugated anti-human secondary IgG. Median fluorescence intensity (MFI) values are indicated. Asterisk (*) indicates surface reactivity to homologous parasite. (**C–E**) Inhibition capacity of the VAR2CSA-specific purified IgG was assessed on three CSA-binding isolates. Black box indicates inhibition activity to the homologous parasite and empty box represent activities against heterologous parasites.

The online version of this article includes the following source data and figure supplement(s) for figure 5:

**Source data 1.** ELISA reactivity of VAR2CSA-specific purified IgG.

**Source data 2.** Inhibition capacity of VAR2CSA-specific purified IgG against M. Camp.

**Source data 3.** Inhibition capacity of VAR2CSA-specific purified IgG against FCR3.

**Source data 4.** Inhibition capacity of VAR2CSA-specific purified IgG against NF54.

**Figure supplement 1.** Reactivity of the post depletion samples to VAR2CSA recombinants and CSA-binding isolates.

strain-transcending naturally acquired anti-adhesion antibodies can be purified on a single recombinant VAR2CSA ectodomain, providing additional evidence that define VAR2CSA as the major target of functional antibodies.

This conclusion is well-supported by previous studies showing that naturally acquired anti-adhesion antibodies in PM target VAR2CSA (*Bigey et al., 2011*; *Doritchamou et al., 2016*), even though other invariant proteins are highly expressed by placental parasites (*Francis et al., 2007*; *Fried et al., 2007*; *Tuikue Ndam et al., 2008*; *Bertin et al., 2013*). Anti-adhesion activity of antibodies targeting these invariant proteins has not been reported thus far, and whether these proteins play a role in parasite binding to CSA is still unclear. Our data demonstrate that the removal of IgG specific to five variants of VAR2CSA ectodomain significantly reduced/abolished IE-surface reactivity and the CSA-binding inhibition activity of total IgG from multigravidae. This observation suggests that antibodies to surface-expressed non-VAR2CSA proteins might have limited or no blocking activity against CSA-binding parasites, although a synergistic effect of these antibodies with anti-VAR2CSA cannot be excluded.

Naturally acquired antibodies of malaria-exposed multigravid women recognize placenta-binding parasites from diverse geographical origins (*Fried and Duffy, 1998*; *Ricke et al., 2000*). One leading hypothesis is that pregnant women acquire antibodies to both cross-reactive and variant-specific epitopes over successive pregnancies and exposure to diverse placenta-binding parasites. We show here that the purification of IgG to VAR2CSA recombinants resulted in loss of IE surface reactivity against the three homologous parasites, and significantly reduced heterologous cross-reactivity of total IgG to maternal isolates. Consequently, the VAR2CSA-purified IgG exhibited strong cross reactivity to all isolates tested in this study, with IgG purified on the first variant demonstrating the highest level of cross-reactivity to diverse CSA-binding isolates. Of note, the isolates under study included WF12/7G8 that expresses the South American VAR2CSA 7G8 allele, in addition to isolates from South East Asia (M Camp), East Africa (M0736), and West Africa (NF54, M2022170, M2001190), thus representing geographically diverse origins. Overall, it may be difficult to estimate the fraction of variant-specific or cross-reactive antibodies contributed by each variant to the measured activities, knowing that each sequence of VAR2CSA could have multiple epitopes that are variably shared with some or all variants (*Benavente et al., 2018*; *Otto et al., 2019*). However, highly cross-reactive antibodies can be purified from a single recombinant VAR2CSA ectodomain, suggesting that immunization with a single variant may prove highly effective.

The full-length VAR2CSA ectodomain-based vaccine approach has been previously investigated in preclinical studies in which antisera raised in rodent models had strong homologous functional activity (*Khunrae et al., 2010*). However, these antibodies generated in rodents did not cross-inhibit different CSA-binding isolates, while exhibiting a broadly strain-transcending surface reactivity to these isolates (*Avril et al., 2011*). In contrast to this observation, our data demonstrate that naturally acquired IgG specific to a single variant and purified from multigravidae sera cross-inhibited multiple isolates. Notably, most of the binding-inhibition activity to CSA-binding parasites was depleted on the first variant and the resulting IgG purified on NF54 (in the first experiment) or M. Camp (in the second experiment) demonstrated strain-transcending blocking activity to all isolates tested in this study. Antibody response to VAR2CSA may differ between species or may differ based on exposure to vaccine antigen or to natural infection. For example, recently published data demonstrated that naturally acquired PfEMP1-specific antibodies are strongly afucosylated, while immunization with a subunit (VAR2CSA) vaccine results in fully fucosylated specific IgG (*Larsen et al., 2021*). Future studies that investigate differences in the host immune response to VAR2CSA in human versus other species will also elucidate any discrepancies between the functional activity of human versus rodent antibody.

Our data support the idea that one VAR2CSA variant might induce antibody with broad functional activity, acknowledging the challenges in developing a full-length VAR2CSA-based vaccine (reviewed in *Doritchamou et al., 2021*), including the possibility that non-functional epitopes may interfere with the optimal functional antibody response. It is also possible that the flanking flexible arm composed of DBL5ε–6ε or DBL5ε–7ε (in some cases) may play a role in masking functional epitopes on the core (NTS-ID3) structure of VAR2CSA (*Doritchamou et al., 2019*; *Ma et al., 2021*). Although recombinant expression of full-length VAR2CSA may not be feasible for manufacturing, novel technologies such as mRNA vaccine platform (reviewed in *Pardi et al., 2018*) should be explored in future studies. To conclude, we report that naturally acquired antibody purified on VAR2CSA ectodomains exhibited strong strain-transcending surface reactivity and blocking activity to CSA-binding isolates.

Interestingly, IgG purified against a single variant captured the bulk of strain-transcending IE reactivity and CSA-binding inhibition activity from total IgG of PM-resistant multigravidae. Therefore, the full-length VAR2CSA ectodomain appears to display both variant-specific and shared epitopes targeted by functional antibodies and as such, would be a credible alternative to the VAR2CSA subunit approach in PM vaccine development. Moreover, cross-reactive antibodies could result from both conserved linear epitopes and conformational (tertiary structure related) epitopes, and the human monoclonal antibody PAM1.4 (*Barfod et al., 2007*) provides strong evidence of the latter. Although such antibodies may contribute to the broadly neutralizing activity of VAR2CSA-specific IgG described in this study, quantifying their relative contribution will inform future PM vaccine development.

# Materials and methods

**Key resources table**

| Reagent type (species) or resource | Designation | Source or reference | Identifiers | Additional information |
|---|---|---|---|---|
| Antibody | Goat anti-Human IgG Fc - Affinity Pure, HRP Conjugate (goat polyclonal) | ImmunoReagents, Inc | GtxHu-004-DHRPX | 1:3000 dilution |
| Antibody | Goat anti-Human IgG Fc Secondary Antibody, PE, eBioscience (goat polyclonal) | Invitrogen, Carlsbad, CA | 12-4998-82 | 1:800 dilution |
| Biological samples (Human) | Plasma | Malaria Research and Training Centre, Bamako, Mali, *Fried et al., 2018* | | |
| Protein, recombinant protein | Full-length VAR2CSA ectodomain, variant NF54 | Patrick E Duffy, *Renn et al., 2021* | | |
| Protein, recombinant protein | Full-length VAR2CSA ectodomain, variant 7G8 | Patrick E Duffy, *Renn et al., 2021* | | |
| Protein, recombinant protein | Full-length VAR2CSA ectodomain, variant FCR3 | Patrick E Duffy, *Renn et al., 2021* | | |
| Protein, recombinant protein | Full-length VAR2CSA ectodomain, variant HB3 | Patrick E Duffy, *Renn et al., 2021* | | |
| Protein, recombinant protein | Full-length VAR2CSA ectodomain, variant M920 | Patrick E Duffy, *Renn et al., 2021* | | |
| Protein, recombinant protein | Full-length VAR2CSA ectodomain, variant M. Camp | Patrick E Duffy, *Renn et al., 2021* | | |
| Protein, recombinant protein | Full-length VAR2CSA ectodomain, variant M200101 | Patrick E Duffy, *Renn et al., 2021* | | |
| Strain, strain background (*Plasmodium falciparum*) | NF54 | Patrick E Duffy, *Doritchamou et al., 2019* | | |
| Strain, strain background (*Plasmodium falciparum*) | FCR3 | Patrick E Duffy, *Doritchamou et al., 2016* | | |
| Strain, strain background (*Plasmodium falciparum*) | M.Camp | Patrick E Duffy | | |
| Strain, strain background (*Plasmodium falciparum*) | M0736 | Patrick E Duffy, *Doritchamou et al., 2016* | | |
| Strain, strain background (*Plasmodium falciparum*) | M2022170 | Patrick E Duffy | | |
| Strain, strain background (*Plasmodium falciparum*) | M2001190 | Patrick E Duffy | | |

*Continued on next page*

*Continued*

| Reagent type (species) or resource | Designation | Source or reference | Identifiers | Additional information |
|---|---|---|---|---|
| Strain, strain background (*Plasmodium falciparum*) | WF12 | Thomas E Wellems, NIH, *Hayton et al., 2008* | | |
| Commercial assay or kit | NHS-Activated Sepharose 4 Fast Flow | GE Healthcare Life Sciences | 17090601 | |
| Chemical compound, drug | SYBR Green I Nucleic Acid Gel Stain - 10,000 X | LifeTechnologies | S7563 | Used at 0.1 X |
| Software, algorithm | Graphpad PRISM 9.0 | GraphPad Software, Inc. | | |
| Software, algorithm | FlowJo 10 | Tree Star, Inc. | | |
| Software, algorithm | R package 'ape' | http://ape-package.ird.fr/ | | |

## Plasma pool, recombinant full-length VAR2CSA and *P. falciparum* cultures

For this study, a plasma pool was prepared using samples from multigravid women participating in the previously described Immuno-epidemiology (IMEP) study (*Fried et al., 2018*). Briefly, pregnant women aged 15–45 years without clinical evidence of chronic or debilitating illness were enrolled from 2010 to 2013 into a longitudinal cohort study of mother-infant pairs conducted in Ouélessébougou, Mali. All participants provided signed informed consent after receiving a study explanation form and oral explanation from a study clinician in their native language. The study protocol and study procedures were approved by the institutional review board of the National Institute of Allergy and Infectious Diseases at the US National Institutes of Health (ClinicalTrials.gov ID NCT01168271), and the Ethics Committee of the Faculty of Medicine, Pharmacy and Dentistry at the University of Bamako, Mali. Aliquots of the plasma pools used in the IgG purification/depletion experiments are fractions of two larger stocks prepared by pooling up to 500 µl of plasma from two batches of 119 and 98 randomly selected multigravid women.

Recombinant full-length VAR2CSA ectodomains of seven genetically diverse *P. falciparum* parasite strains (NF54, 7G8, FCR3, HB3, M. Camp, M920, and M200101) (*Figure 2—figure supplement 4*) were expressed in a mammalian expression system as in a previous study (*Renn et al., 2021*). Three CSA-binding maternal isolates (M2022170, M2001190, and M0736) expressing VAR2CSA as well as four lab-strains (NF54, FCR3, M. Camp and WF12/7G8, a progeny of 7G8 X GB4 (*Hayton et al., 2008*) that encodes the 7G8 VAR2CSA variant) selected to bind CSA (*Supplementary file 1*) were maintained in culture for the functional assays.

## Purification of IgG specific to full-length VAR2CSA ectodomains

Total IgG was initially purified from 4 mL of the multigravidae plasma pool using Protein G Sepharose 4 Fast Flow resin (GE Healthcare Life Sciences) according to the manufacturer's instructions. For VAR2CSA-specific IgG purification and depletion, an antigen-specific affinity column was prepared as previously described (*Doritchamou et al., 2016*) for each VAR2CSA variant. Briefly, each ectodomain variant was chemically cross-linked to N-hydroxysuccinimide (NHS)-activated Sepharose beads (GE Healthcare Life Sciences) according to the manufacturer's instructions. IgG specific to five VAR2CSA variants (NF54, 7G8, FCR3, HB3, and MC) were sequentially removed from the total IgG by passage through the affinity column and by applying the flow-through sample from one column to the next column until all specificities were depleted (*Figure 1*: Flow chart). IgG purification was performed repeatedly on each VAR2CSA antigen to ensure complete depletion of the specific IgG. The purified IgG were eluted with IgG Elution buffer, pH 2.8 (Invitrogen, Carlsbad, CA), neutralized with 2 M Tris buffer pH 9.0, and dialyzed into PBS pH 7.4. Variable number of passages were needed for full depletion of NF54 (x 3), 7G8 (x 3), FCR3 (x 2), HB3 (x 1) and M. Camp (x 1) specific IgG. The different aliquots of purified IgG were quantified using a Nanodrop (ND2000, Thermo Fisher Scientific, Waltham, MA).

## Reactivity of the naturally acquired IgG to VAR2CSA recombinants and VAR2CSA-expressing isolates

Total IgG (before and after each variant-specific depletion) or VAR2CSA-specific IgG purified on each the five ectodomain variants was assessed by ELISA as previously described (*Doritchamou et al.,*

*2019*). Briefly, recombinant VAR2CSA proteins were coated at 1 µg/mL and incubated overnight at 4 °C. Plates were blocked at room temperature (RT) for 2 hr (hr) and 100 µL of total IgG at 10 µg/mL and VAR2CSA-specific IgG at 0.1 µg/mL were added in duplicate wells for 1 hr at RT. The plates were washed and 100 µL of 1:3000 diluted HRP-conjugated anti-human IgG antibody (ImmunoReagents, Inc) added to each well for 1 hr incubation at RT followed by a final wash. One hundred microliters TMB (SeraCare) was added to each well for 10 min of incubation in dark at RT and the reaction was stopped by adding an equal volume of Stop Solution (SeraCare). Optical density (OD) values at 450 nm were acquired using the MultiskanFC (Thermo Fisher) plate reader.

IgG reactivity to native VAR2CSA expressed on IE surface was assessed by flow cytometry using 6 isolates of *P. falciparum* as previously described (*Doritchamou et al., 2016*). Briefly, enriched mature trophozoite/schizont stages of IE were incubated with total IgG (at 100 µg/mL) or the antigen-specific IgG (at 1 µg/mL), washed, and bound IgG labeled with PE-conjugated anti-human IgG (Invitrogen, Carlsbad, CA) in a buffer containing 0.1% SYBR (LifeTechnologies). IgG-labeled IE were quantified using an LSRII flow cytometer (BD Biosciences, San Jose, CA) and analyzed in FlowJo 10 (Tree Star, Inc). The median fluorescence intensity (MFI) was determined and background intensity from a well containing IE without immune IgG was subtracted from the MFI value detected for each test IgG.

## Inhibition of IE binding to CSA

The CSA-binding inhibition activity of the purified IgG was evaluated in a static binding inhibition assay using immobilized CSPG receptor as previously described (*Doritchamou et al., 2016*). Briefly, the proteoglycan decorin (Sigma) at 2 µg/mL in 1 X PBS was coated as 15 µL spots on a 100 × 15 mm Petri dish (Falcon 351029) by overnight incubation at 4 °C in a humid chamber. The spots were blocked with 3% BSA in 1 X PBS at 37 °C for 30 min. Before binding assay, enriched mature trophozoite/schizont stages of IE from different *P. falciparum* strains were adjusted to 20% parasite density at 0.5% hematocrit and incubated with total IgG (at 1, 2, and 4 mg/mL) or VAR2CSA-specific IgG (at 0.1 mg/mL) for 30 min at 37 °C. IE suspended in IgG solution were then allowed to bind duplicate receptor spots for 30 min at RT. Unbound IE were washed away and bound IE were fixed, stained and quantified by microscopy. The percentage of inhibition was calculated relative to the wells containing IE without IgG.

## Acknowledgements

This work was supported by the Intramural Research Program of the National Institute of Allergy and Infectious Diseases, National Institutes of Health. The authors thank J Patrick Gorres for editing the manuscript and Robert Morrison for bioinformatic analysis. We are grateful to women in Ouélessébougou, Mali for participation in the IMEP study. We also thank Thomas E Wellems for generously gifting us the WF12 parasite clone.

## Additional information

### Funding

| Funder | Grant reference number | Author |
|---|---|---|
| Division of Intramural Research, National Institute of Allergy and Infectious Diseases | | Justin YA Doritchamou Jonathan P Renn Bethany Jenkins Michal Fried Patrick E Duffy |
| National Institutes of Health | | Justin YA Doritchamou Jonathan P Renn Bethany Jenkins Michal Fried Patrick E Duffy |

The funders had no role in study design, data collection and interpretation, or the decision to submit the work for publication.

## Author contributions
Justin YA Doritchamou, Conceptualization, Data curation, Formal analysis, Investigation, Supervision, Validation, Visualization, Writing - original draft; Jonathan P Renn, Conceptualization, Investigation, Writing – review and editing; Bethany Jenkins, Formal analysis, Investigation, Writing – review and editing; Almahamoudou Mahamar, Writing – review and editing, Conducted the clinical study in Mali and provided samples used in the study; Alassane Dicko, Writing – review and editing, Conducted the clinical study in Mali and provided samples used in the study; Michal Fried, Funding acquisition, Supervision, Writing – review and editing; Patrick E Duffy, Conceptualization, Funding acquisition, Supervision, Writing – review and editing

## Author ORCIDs
Justin YA Doritchamou (iD) http://orcid.org/0000-0002-4589-7216
Patrick E Duffy (iD) http://orcid.org/0000-0002-4483-5005

## Ethics
Human subjects: For this study, a plasma pool was prepared using samples from multigravid women participating in the previously described Immuno-epidemiology (IMEP) study (Fried et al., 2018). Briefly, pregnant women aged 15-45 years without clinical evidence of chronic or debilitating illness were enrolled from 2010-2013 into a longitudinal cohort study of mother-infant pairs conducted in Ouélessébougou, Mali. All participants provided signed informed consent after receiving a study explanation form and oral explanation from a study clinician in their native language. The study protocol and study procedures were approved by the institutional review board of the National Institute of Allergy and Infectious Diseases at the US National Institutes of Health (ClinicalTrials.gov ID NCT01168271), and the Ethics Committee of the Faculty of Medicine, Pharmacy and Dentistry at the University of Bamako, Mali.

## Decision letter and Author response
Decision letter https://doi.org/10.7554/eLife.76264.sa1
Author response https://doi.org/10.7554/eLife.76264.sa2

---

# Additional files

## Supplementary files
• Supplementary file 1. CSA-binding level of the isolates.
• Transparent reporting form

## Data availability
All data generated or analysed during this study are included in the manuscript and supporting file; source data files have been provided for all figures.

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
