## [Decision Letter]

**Decision letter after peer review:**

[Editors’ note: the authors submitted for reconsideration following the decision after peer review. What follows is the decision letter after the first round of review.]

Thank you for resubmitting the paper entitled "A single full-length VAR2CSA ectodomain variant purifies broadly neutralizing antibodies against placental malaria isolates" for further consideration by *eLife*. Your revised article has been reviewed by three peer reviewers, one of whom is a member of our Board of Reviewing Editors, and the evaluation has been overseen by a Senior Editor. We are sorry to say that we have decided that this submission will not be considered further for publication by *eLife*.

While all reviewers found your work of interesting to the community, they also agreed that the findings reported are too preliminary to fully justify the conclusions drawn. In particular, the lack of permutated depletion experiments was considering a critical shortcoming, and that substantial additional experimentation will be necessary to alleviate this concern and the other concerns raised and detailed below.

*Reviewer #1:*

This is a follow-up of a similar study by the same authors (Doritchamou, 2016). While they previously used recombinant single- and double-domain constructs of VAR2CSA to affinity-purify IgG from placental malaria-exposed multigravidae, they now employ full-length VAR2CSA ectodomain constructs. In contrast to the findings from their earlier study, the authors now report that affinity purification of IgG on full-length VAR2CSA yielded IgG that contained broadly neutralizing (adhesion-blocking) activity. The study reports what appears to be essentially a single sequential experiment that is consistent with the authors' interpretation, but in my opinion falls somewhat short of fully justifying it.

Reactivity and neutralization data of experiments all variants and single variant-depleted and single variant-purified IgG would strengthen the study markedly. Furthermore, it would be interesting to see results from experiments where the order of the sequential depletion was permutated. Finally, the authors should present absolute adhesion and adhesion inhibition data, as normalized data can be quite misleading (e.g., they can produce impressive adhesion inhibition data for a parasite line that shows almost no adhesiveness initially).

Specific comments:

– L. 15 ("…it is still unclear… epitopes"): The authors themselves have previously reported data supporting the existence of broadly cross-reactive IgG acquired in response to natural infection (Barfod et al. 2010).

– L. 25-8: I find this conclusion too categorical, considered the limited evidence (the above-mentioned "single" experiment and the fact that only a few maternal isolates – from the same study site and temporally related? – were tested).

– L. 41-2: Expression of VAR2CSA was not formally demonstrated in Fried 1998 or Ricke 2000 as both studies predate the discovery of VAR2CSA.

– L. 46-8 could deserve referencing – either original studies or a recent review (e.g., Hviid and Jensen 2015).

– L. 51-4: Suggest adding Wang et al. Nat Commun 2021.

– L. 54-8: Nine references seems a bit excessive here.

– L. 153-5 ("Although… NF54-IgG"): Why did the authors not test for IgG reactivity against native 7G8-VAR2CSA and HB3-VAR2CSA on the surface of IEs?

– L. 155ff ("As with ELISA… antigen"): How does this sentence fit with the fact that reactivity against 7G8-IEs was not tested?

*Reviewer #2:*

This work builds on both work from this team suggesting that antibodies to individual VAR2CSA domains do not show broad cross reactive neutralising activity, and on two phase 1 trials of vaccines based on the DBL2 domain of VAR2CSA and flanking sequences, which showed limited heterologous protection in in vitro studies. This work has led to the idea that it might be necessary to include a cocktail of key variants to produce a successful vaccine. The current authors instead investigated whether antibodies purified using the whole protein ectodomain were markers of potential heterologous protection, and find that this is the case.

Generally the experiments presented and the interpretations of the findings appear logical and consistent. The figures show that depletion of antibodies using one VAR2CSA variant decreases antibody reactivity to other variants by ELISA and that the more the plasma is depleted, the greater the loss of antibody reactivity. Broadly similar data are shown for the ability of plasma to inhibit adhesion of infected erythrocytes (IEs) to CSA, and for ability to recognise the IE surface using flow cytometry. The validation of their ELISA findings with live parasites is a strength of this study.

The results are consistent with the idea that immunity to a single VAR2CSA variant might be associated with a fairly broad range of cross-reactive antibody to different VAR2CSA isolates that might broadly protect against placental malaria. It is not presently clear whether production of such an antigen at scale for use in vaccines is a realistic possibility, and this challenge might be mentioned in the discussion. It is also not clear whether immunisation with a single variant would result cross reactive immunity (recognition of an antigen by antibodies being different to an antigens ability to elicit antibodies). The results contrast with the animal immunisation studies mentioned in the discussion; however, the authors do not clearly discuss why this may be the case.

Specific suggestions

Line 84-87: should this be "that are absent in 'individual' vaccine fragments"? As this study does not confirm that all vaccine fragments together do not contain epitopes.

Lines 105 and 113-114: The data presented in figures 2 are different to those in supplementary figure 1. Therefore, the results from Figures in the main file and supplement should be described separately.

Line 263-7: We are missing details of how the ectodomains were expressed. There is a reference to Renn et al., JEM 20210848P which may be a submitted manuscript. This information is important in understanding how the protein expression was evaluated for conformation and folding, as it appears tertiary or higher-level structural considerations are important for understanding the difference between this study and previous efforts using single domains.

Discussion: Could the authors comment on whether it is that they have used a more complete protein or whether it is the tertiary structure which results in the cross-reactive antibodies? Would the use of for example 6 fragments (making up the whole VAR2CSA) give a similar result?

Figure 2 A: what do the brackets and asterisk refer to?

Figure 2 C: There is a comment about "histogram without IgG" – what does this refer to? Does it refer to B rather than C?

Figure 2 D may be redundant as it can be derived from figure 2C.

Figure 3 X axis should not be VAR2CSA, it is antibody reactivity to IEs.

Line 179- should this be 'heterologous parasites' not homologous as it is clear that other purifications did indeed capture IgG that could block activity against heterologous parasite strains (though not at the same level as that of the first depletion).

Line 202: what is an 'invariant protein'?

Methods:

Line 279-281 More detail of the elution buffer is also needed as low pH can cause changes to antibody conformation that might alter protein binding.

Lines 290,298-9 please provide details of the reagents used including detection antibodies and SYBR.

How many times did plasma need to be passed through column before activity was deemed fully depleted?

*Reviewer #3:*

1) It has been shown in a previous work (in particular by the first author of this work), that despite exposure to several field *P. falciparum* isolates, multigravid women could nevertheless be susceptible to placental infections by certain rare parasite variants (resembling in their var2csa sequence to strain WRO80). The authors should have discussed their findings in relation to these very relevant observations from this previous study?

2) Including laboratory strains 7G8 and HB3 in the cellular testings (surface reactivity and binding inhibition) would have helped to substantiate the authors' conclusions. The same applies to parasite isolates M920 and M200101 from which the FL var2csa recombinants have been produced.

3) The order of filtration of the MG IgG pool on the antigens during depletion / purification steps seems decisive. If this order had been different would the authors have expected the same activity pattern of the purified antibodies? It would have been interesting to reproduce this experiment by reversing this order. Carrying out this experiment in my opinion would have further strengthened the study and better reflect the title chosen by the authors. One can wonder whether all the VAR2CSA variants used in this study would be functionally comparable in this operation, if not the title of this study would be better suited to IgG purified on the VAR2CSA variant of NF54, as the depletion / purification activity on this protein (first in the series) presents the least bias compared to the others used afterwards.

4) Despite these comments which could only further help improve this work if addressed, the content is already of definite interest and may be published or with some few modifications.

5) Adding a figure (even as a supplementary) that shows the dispersion or phylogenic relation between the 7 VAR2CSA sequences used in this work would have been a good added value, as this will inform about their relationship with the main variants (3D7, FCR3, 7G8, WRO80) and help to further interpret the results.

6) The total IgGs purified from the MG pool show a strong reactivity against the 7 recombinants VAR2CSA used as well as on the native protein of the cultured parasites. This surface reactivity of the pre-depletion IgG pool is surprisingly less on the strain NF54 (Figure 2B). In Figure 3 the affinity-purified IgGs on the NF54 antigen (the first in the chain of depletion/purification) also show strong reactivity against the 7 recombinants used as well as on the native protein of all the 6 parasites cultured, including NF54. The authors should explain or discuss this weak recognition of the pre-depletion pool on the NF54 parasite which contrasts with the inhibitory properties observed (Figure 2C)?

7) Authors should describe how the plasma pool was prepared. It would be helpful to clarify how many multigravid women have been used as donors? If they were selected it would be necessary to clarify on which criteria and if not why.

---

## [Author Response]

[Editors’ note: the authors resubmitted a revised version of the paper for consideration. What follows is the authors’ response to the first round of review.]

Reviewer #1:This is a follow-up of a similar study by the same authors (Doritchamou, 2016). While they previously used recombinant single- and double-domain constructs of VAR2CSA to affinity-purify IgG from placental malaria-exposed multigravidae, they now employ full-length VAR2CSA ectodomain constructs. In contrast to the findings from their earlier study, the authors now report that affinity purification of IgG on full-length VAR2CSA yielded IgG that contained broadly neutralizing (adhesion-blocking) activity. The study reports what appears to be essentially a single sequential experiment that is consistent with the authors' interpretation, but in my opinion falls somewhat short of fully justifying it.

In an additional set of experiments, we have now reproduced our findings in studies in which we changed the order of VAR2CSA antigens used for the sequential purification/depletion of IgG. Our data confirm that antibody purified on the first antigen exhibits broadly neutralizing activity, and this is not dependent on the VAR2CSA variant used for initial purification.

Reactivity and neutralization data of experiments all variants and single variant-depleted and single variant-purified IgG would strengthen the study markedly. Furthermore, it would be interesting to see results from experiments where the order of the sequential depletion was permutated. Finally, the authors should present absolute adhesion and adhesion inhibition data, as normalized data can be quite misleading (e.g., they can produce impressive adhesion inhibition data for a parasite line that shows almost no adhesiveness initially).

We thank the reviewer for these suggestions. Unfortunately, our lab does not have both 7G8 and HB3 isolates. However, in this revised manuscript, we have included data generated on WF12, a progeny of 7G8 X GB4 whose genome encodes the 7G8 VAR2CSA allele (Hayton et al., Cell Host Microbe 2008). We have provided detailed information on WF12 in Supplemental Figure S2. We have also permutated the order of antigens in a second sequential depletion experiment using full-length VAR2CSA (FV2) representing Malayan Camp (MCamp), with FCR3 and NF54 variants (in that order) for purification/depletion. In our additional data, we show that IgG purified on FV2 MCamp had the highest cross-reactivity and cross-inhibitory activity against recombinant FV2 variants as well as against native antigen expressed by MCamp, FCR3 and NF54 parasites (Figure 5). This finding is consistent with the pattern observed with IgG purified on NF54 variant in the first sequential depletion experiment in our original manuscript. We have provided average CSA binding level for the isolates used in the BIA experiments as Table S1.

– L. 25-8: I find this conclusion too categorical, considered the limited evidence (the above-mentioned "single" experiment and the fact that only a few maternal isolates – from the same study site and temporally related? – were tested).

As noted above, we have now generated new data, using a different order of FV2 variants for antibody purification/depletion on the multigravidae plasma pool, and depleting initially with a west African variant (NF54) in the first set of experiments and in the second set of experiments depleting initially with southeast Asian variant (M Camp). We have incorporated an additional parasite (that expresses the South American VAR2CSA 7G8 allele), in addition to the isolates from SE Asia (M Camp), east Africa (M0736) and west Africa (NF54, M2022170, M2001190) reported in the original manuscript. We believe that the additional data from the second sequential depletion experiment and the expanded set of parasites support our conclusions.

– L. 41-2: Expression of VAR2CSA was not formally demonstrated in Fried 1998 or Ricke 2000 as both studies predate the discovery of VAR2CSA.

These two references referred to antibodies that bind to placental IE. The sentence has been modified for clarity (lines 43-44).

– L. 46-8 could deserve referencing – either original studies or a recent review (e.g., Hviid and Jensen 2015).

As suggested, references have been added (lines 50-51).

– L. 51-4: Suggest adding Wang et al. Nat Commun 2021.

The suggested reference has been added (line 57).

– L. 153-5 ("Although… NF54-IgG"): Why did the authors not test for IgG reactivity against native 7G8-VAR2CSA and HB3-VAR2CSA on the surface of IEs?

As indicated above, our lab does not have both 7G8 and HB3 isolates. We have now tested the purified IgG on WF12, a progeny of 7G8 X GB4 that expresses the same native VAR2CSA as South American 7G8 parasite.

Reviewer #2:This work builds on both work from this team suggesting that antibodies to individual VAR2CSA domains do not show broad cross reactive neutralising activity, and on two phase 1 trials of vaccines based on the DBL2 domain of VAR2CSA and flanking sequences, which showed limited heterologous protection in in vitro studies. This work has led to the idea that it might be necessary to include a cocktail of key variants to produce a successful vaccine. The current authors instead investigated whether antibodies purified using the whole protein ectodomain were markers of potential heterologous protection, and find that this is the case.Generally the experiments presented and the interpretations of the findings appear logical and consistent. The figures show that depletion of antibodies using one VAR2CSA variant decreases antibody reactivity to other variants by ELISA and that the more the plasma is depleted, the greater the loss of antibody reactivity. Broadly similar data are shown for the ability of plasma to inhibit adhesion of infected erythrocytes (IEs) to CSA, and for ability to recognise the IE surface using flow cytometry. The validation of their ELISA findings with live parasites is a strength of this study.The results are consistent with the idea that immunity to a single VAR2CSA variant might be associated with a fairly broad range of cross-reactive antibody to different VAR2CSA isolates that might broadly protect against placental malaria. It is not presently clear whether production of such an antigen at scale for use in vaccines is a realistic possibility, and this challenge might be mentioned in the discussion. It is also not clear whether immunisation with a single variant would result cross reactive immunity (recognition of an antigen by antibodies being different to an antigens ability to elicit antibodies). The results contrast with the animal immunisation studies mentioned in the discussion; however, the authors do not clearly discuss why this may be the case.

We thank the reviewer for these comments, and we have made revisions and clarifications to our manuscript to address them. We share the reviewer’s concern about the realistic possibility to produce such a large antigen at scale for use in vaccines. However, we believe the mRNA vaccine platform (among other advanced technologies) could be a feasible approach to explore and have discussed this in the revised manuscript (Lines 272-275). We also recognized the limitation of antibodies induced against FV2 in animals to display broadly neutralizing activity, similar to what is seen in naturally acquired immunity. It is possible that differences in host responses and context of the antigen may alter quality of immunity. For example, recently published data demonstrated that naturally acquired PfEMP1-specific antibodies are strongly afucosylated, while immunization with a subunit (VAR2CSA) vaccine results in fully fucosylated specific IgG (Larsen et al., 2021). This study provides new understanding of key differences in antibody features between natural- and vaccine-induced immunity. The Discussion section has been amended with this new observation in the revised manuscript (lines 258-264).

Lines 105 and 113-114: The data presented in figures 2 are different to those in supplementary figure 1. Therefore, the results from Figures in the main file and supplement should be described separately.

Figure 2 and Supplementary Figure S1 are both used to demonstrate depletion of reactivity to VAR2CSA antigens and parasites from total IgG. However, Supplementary Figure S1 shows reactivity after each individual antigen-specific IgG purification/depletion, while Figure 2 only displays the difference between Pre-depletion IgG versus IgG post-depletion by all variants used.

Line 263-7 We are missing details of how the ectodomains were expressed. There is a reference to Renn et al., JEM 20210848P which may be a submitted manuscript. This information is important in understanding how the protein expression was evaluated for conformation and folding, as it appears tertiary or higher-level structural considerations are important for understanding the difference between this study and previous efforts using single domains.

That is an interesting point, and the reference has been updated in this revised version of the manuscript. We cite our new publication (Renn et al., Comms Bio 2021, DOI : 10.1038/s42003-021-02787-7) that provides a detailed description of the expression and functional characterization of full length VAR2CSA ectodomain variants (lines 301-306).

Discussion: Could the authors comment on whether it is that they have used a more complete protein or whether it is the tertiary structure which results in the cross-reactive antibodies? Would the use of for example 6 fragments (making up the whole VAR2CSA) give a similar result?

At the current state of understanding, it appears that cross-reactive antibodies could result from both conserved linear epitopes and conformational (tertiary structure related) epitopes. The strain-transcending human monoclonal antibody PAM1.4 provides strong evidence of a strain-transcending antibody targeting conformational epitopes on full-length VAR2CSA (Barfod et al., 2007). Although such antibodies may be contributing to the broadly neutralizing activities of PM IgG described in this study, quantifying their level of contribution will certainly provide crucial data to inform PM vaccine development. Similarly, whether the use of the 6 individual domains (making up the entirety of the VAR2CSA ectodomain) will give a similar result remains to be explored. The Discussion has been amended accordingly (lines 281-286)

Line 202: what is an 'invariant protein'?

Here, an “invariant protein” refers to a highly conserved protein across different variants of *P. falciparum* encoded by a single-copy gene. For example, we have a new publication that describes such an invariant protein on the surface of CSA-binding IE (Keitany et al., J Infect Dis. doi: 10.1093/infdis/jiab550).

Reviewer #3:1) It has been shown in a previous work (in particular by the first author of this work), that despite exposure to several field *P. falciparum* isolates, multigravid women could nevertheless be susceptible to placental infections by certain rare parasite variants (resembling in their var2csa sequence to strain WRO80). The authors should have discussed their findings in relation to these very relevant observations from this previous study?

We thank the reviewer for this comment. We have indicated in this manuscript and Renn et al., 2021 that the VAR2CSA sequence of M920 has a similar ID1 motif as WR80. Our data here show that IgG purified on the first FV2 variant significantly cross-reacted with the M920 variant of VAR2CSA. Unfortunately, the M920 isolate was not available to us to generate any surface reactivity and binding inhibition data. At this point, we believe that our interpretation should be limited to the observed ELISA cross-reactivity to a WR80-like VAR2CSA.

2) Including laboratory strains 7G8 and HB3 in the cellular testings (surface reactivity and binding inhibition) would have helped to substantiate the authors' conclusions. The same applies to parasite isolates M920 and M200101 from which the FL var2csa recombinants have been produced.

We do not have 7G8 and HB3 strains but have included data generated on WF12 (a progeny of 7G8 X GB4 that expresses the same VAR2CSA sequences as 7G8 (Hayton et al., Cell Host Microbe 2008)) in this revised version of the manuscript. We believe that data generated from maternal isolates M2022170 (Mali), M2001190 (Mali) and M0736 (Tanzania, for which no VAR2CSA) protein was produced, have provided reasonably similar findings (substantial reduction of reactivities and inhibition after depletion of IgG specific to multiple variants of VAR2CSA) as those from M920 (parasite not available) and M200101 (fails to maintain expression of VAR2CSA in culture) would have provided.

3) The order of filtration of the MG IgG pool on the antigens during depletion / purification steps seems decisive. If this order had been different would the authors have expected the same activity pattern of the purified antibodies? It would have been interesting to reproduce this experiment by reversing this order. Carrying out this experiment in my opinion would have further strengthened the study and better reflect the title chosen by the authors. One can wonder whether all the VAR2CSA variants used in this study would be functionally comparable in this operation, if not the title of this study would be better suited to IgG purified on the VAR2CSA variant of NF54, as the depletion / purification activity on this protein (first in the series) presents the least bias compared to the others used afterwards.

We thank the reviewer for this suggestion and have carried on generating similar data from a second series of sequential FV2 IgG purifications on MCamp, FCR3 and NF54 variants, in that order. This time, IgG first purified on MCamp (SE Asia variant) displayed the highest level of cross-reactivities and cross-inhibition activity, similar to what was seen in our original experiment with IgG purified first on NF54 (west African variant). These new data support the idea that different full-length VAR2CSA variants used as a first antigen to deplete/purify IgG from multigravidae will produce comparable functional activities as seen with NF54 and MCamp.

5) Adding a figure (even as a supplementary) that shows the dispersion or phylogenic relation between the 7 VAR2CSA sequences used in this work would have been a good added value, as this will inform about their relationship with the main variants (3D7, FCR3, 7G8, WRO80) and help to further interpret the results.

A larger phylogenic analysis of VAR2CSA sequences including the 7 variants that were expressed is reported in our newly published paper that described the expression of these FV2 recombinants (Renn et al., 2021). We have also provided a phylogenetic tree with all the 7 variants of FV2 used in this study as Supplemental Figure S7. The NTS-DBL6 version of M200101 (NTS-DBL7) VAR2CSA sequence has been also included here as reference.

6) The total IgGs purified from the MG pool show a strong reactivity against the 7 recombinants VAR2CSA used as well as on the native protein of the cultured parasites. This surface reactivity of the pre-depletion IgG pool is surprisingly less on the strain NF54 (Figure 2B). In Figure 3 the affinity-purified IgGs on the NF54 antigen (the first in the chain of depletion/purification) also show strong reactivity against the 7 recombinants used as well as on the native protein of all the 6 parasites cultured, including NF54. The authors should explain or discuss this weak recognition of the pre-depletion pool on the NF54 parasite which contrasts with the inhibitory properties observed (Figure 2C)?

The lower reactivity of the pre-depletion total IgG (used at 100ug/mL) compared to the antigen-specific IgG (used at 1ug/mL) could be explained by the low concentration of VAR2CSA-specific IgG in the total IgG. Indeed, based on the antibody yield in Table 1, NF54-specific IgG might represent 1/400 fraction of the total IgG. In a tentative extrapolation of the Flow data, a level of surface reactivity similar to that of NF54-specific IgG might be obtained if pre-depletion total IgG was used at 400ug/mL.

7) Authors should describe how the plasma pool was prepared. It would be helpful to clarify how many multigravid women have been used as donors? If they were selected it would be necessary to clarify on which criteria and if not why.

The plasma pools used in this study are fractions of two larger stocks prepared by pooling up to 500ul of plasma from 119 (first purification/depletion experiment) and 98 (second purification/depletion experiment) randomly selected multigravid women. We have now provided this detail in the revised manuscript.